# Flanker Task Performance in Isolated Dystonia (Blepharospasm): A Focus on Sequential Effects

**DOI:** 10.3390/brainsci10020076

**Published:** 2020-02-01

**Authors:** Max Pekrul, Caroline Seer, Florian Lange, Dirk Dressler, Bruno Kopp

**Affiliations:** 1Department of Neurology, Hannover Medical School, Carl-Neuberg-Straße 1, 30625 Hannover, Germany; caroline.seer@kuleuven.be (C.S.); florian.lange@kuleuven.be (F.L.); dressler.dirk@mh-hannover.de (D.D.); kopp.bruno@mh-hannover.de (B.K.); 2Movement Control & Neuroplasticity Research Group, Department of Movement Sciences, KU Leuven, Tervuursevest 101, 3001 Leuven, Belgium; 3LBI—KU Leuven Brain Institute, KU Leuven, 3000 Leuven, Belgium; 4Behavioral Engineering Research Group, KU Leuven, Naamsestraat 69, 3000 Leuven, Belgium; 5Movement Disorders Section, Department of Neurology, Hannover Medical School, Carl-Neuberg-Straße 1, 30625 Hannover, Germany

**Keywords:** dystonia, blepharospasm, hemifacial spasm, executive function, flanker task, congruency sequence effect, response sequence effect

## Abstract

Isolated dystonia manifests with involuntary muscle hyperactivity, but the extent of cognitive impairment remains controversial. We examined the executive functions in blepharospasm while accounting for motor symptom-related distractions as a factor often limiting the interpretability of neuropsychological studies in dystonia. Our control group comprised of patients with hemifacial spasm, which is a condition producing similar motor symptoms without any central nervous system pathology. Nineteen patients with blepharospasm and 22 patients with hemifacial spasm completed a flanker task. Stimulus congruency on the current trial, on the preceding trial, and a response sequence served as independent variables. We analyzed the response time and accuracy. Gross overall group differences were not discernible. While congruency, congruency sequence, and response sequence exerted the expected effects, no group differences emerged with regard to these variables. A difference between patients with blepharospasm and those with hemifacial spasm consisted in longer reaction times when responses had to be repeated following stimulus incongruency on the preceding trial. We conclude that patients with blepharospasm seem to have difficulties in repeating their responses when incongruency on preceding trials interferes with habit formation or other forms of fast routes to action. Our specific finding may provide an opportunity to study altered basal ganglia plasticity in focal dystonia.

## 1. Introduction

Isolated dystonia is a neurological condition of impaired motor control resulting in involuntary muscle contractions, twisting movements, and sustained or intermittent abnormal postures [1]. There is emerging evidence that adult onset focal dystonia is not limited to motor impairment but may also affect cognitive functions [2,3,4,5,6,7,8]. Other studies did not report cognitive impairments in patients with focal dystonia [9,10,11]. This disagreement suggests that cognitive impairment in patients with focal dystonia, if present, might be rather specific, i.e., restricted to a narrow range of cognitive functions.

Structural and/or functional changes in focal dystonia may be present in the basal ganglia (BG). The BG hypothesis of focal dystonia was derived from studies focusing on these subcortical structures [12,13,14] including BG lesioning [15,16] and deep brain stimulation (DBS) studies [17,18]. This BG view was broadened [19,20] such that focal dystonia has rather been regarded as a network disease [21,22,23,24,25,26,27,28]. These latter studies identified structural and functional changes in additional brain regions, and they proposed a crucial role of the BG, cerebellum, neocortex, and their manifold interconnecting pathways in the pathophysiology of focal dystonia. 

Blepharospasm (BSP) is a manifestation of isolated focal dystonia with onset in adulthood that affects the eyelid muscles (Mm. orbicularis oculi) bilaterally [29]. Studies of cognitive changes in BSP are often limited by motor symptom-related confounders. When compared to healthy control participants, patients’ performance deficits on cognitive tasks might be due to cognitive impairment or the distracting impact of motor symptoms during cognitive testing [10,30,31]. In order to address this problem, the control group in our study consisted of patients suffering from hemifacial spasm (HFS). Although HFS (involuntary unilateral eye closure) is associated with similar motor symptoms as BSP (involuntary bilateral eye closure), its etiology is entirely localized in the peripheral nervous system. Comparing cognitive test performance between patients with BSP and patients with HFS thus allowed us to study the contribution of BSP-related central nervous impairment to cognitive impairment while controlling for motor symptom-related dysfunctions.

A link between blepharospasm and executive functions is suggested by the putative involvement of fronto-striatal loops in the pathogenesis of typical motor symptoms as well as in the pathogenesis of executive dysfunctions [32]. Possible cognitive impairments of patients with BSP might be assessable with a task addressing executive functions (EF; see [33] for an overview). EF are a multi-faceted construct. Miyake and coworkers identified three EF facets by means of latent variable analysis: set shifting, information updating, and response inhibition [34]. Response inhibition manifests itself in tasks requiring responses that are in conflict with automatically primed or dominant responses.

The paradigm applied in this study is a modified version of the Eriksen flanker task [35,36], which is supposed to provide behavioral measures of response inhibition. In the modified task version, stimuli consist of a central target arrowhead, which is flanked by two distracting arrowheads. The flanker stimuli can facilitate the reaction to the target stimulus by pointing in the same direction (congruent condition). Alternatively, the flanker stimuli can interfere with the reaction to the target stimulus by pointing in the opposite direction (incongruent condition), potentially inducing cognitive conflict (see [37] for an overview). It is the latter case that putatively requires the inhibition of erroneous responses that are primed by incongruent flankers. Typical behavioral measures on the flanker task are reaction times (RT, speed) and error rates (ER, accuracy). Responses to congruent stimuli are faster and more accurate than responses to incongruent stimuli, which are referred to as the congruency effect. 

The sequence of stimuli across successive trials plays a crucial role for adjustments of task performance and is known as the congruency sequence effect (CSE). The CSE can be most easily understood as a modulation of the congruency effect. Thus, following congruent trials, congruency effects are typically more pronounced than after incongruent trials [38]. The CSE may reflect attentional adjustments, with a widened spatial extension of attention following congruency (hence, the more pronounced congruency effects on the following trial) and a narrowed spatial extension of attention following incongruency (hence, the less pronounced congruency effects on the following trial). Another factor that contributes to the sequential modulation of task performance is response sequence. The current trial can either require a repetition of the previous response (e.g., a left-hand response followed by a left-hand response) or a response alternation (e.g., a right-hand response followed by a left-hand response). In simple serial RT tasks, response repetitions are typically associated with faster and more accurate responding. However, alterations of task features, or the presentation of very long response–stimulus intervals (RSI), may lead to the emergence of repetition RT costs [39,40,41,42].

We examined whether (and to what extent) behavioral indices of response inhibition and their sequential modulation by stimulus sequence and by response sequence are altered in patients with BSP when compared to patients with HFS on the flanker task. Patients with idiopathic Parkinson’s disease (PD)—a disease with a known degeneration of dopaminergic projections from the substantia nigra to the dorsal striatum—showed evidence for a strongly reduced CSE in a very similar task design [43]. Thus, the present study provides the opportunity to examine the sequential modulation of task performance on a flanker task in patients with BSP—in particular, the effects of stimulus sequence and response sequence—and to compare congruency and sequential effects directly with those that can be observed in patients with HFS. Furthermore, these congruency and sequential effects can be indirectly compared with those that had been observed in patients with PD in previous research. Previous flanker studies in dystonia have not been published. Therefore, our study is rather exploratory.

## 2. Materials and Methods

### 2.1. Participants

Nineteen patients with BSP were recruited from the Movement Disorders Section of the Department of Neurology at Hannover Medical School. All of them were diagnosed by an experienced neurologist (DD) and had normal or corrected-to-normal vision. The patients did not have any psychiatric or neurological disorder except for BSP. One patient was excluded from analysis due to prolonged reaction times (> 3 SD above the sample mean). The final patient sample (*N* = 18, 10 female, 8 male) consisted of 14 right-handed and 3 left-handed participants (this information could not be obtained from one patient). Two patients were diagnosed with additional oromandibular dystonia. 

Twenty-two participants with HFS served as the control group. They were recruited from the same outpatient clinic and diagnosed by the same neurologist (DD). All controls had normal or corrected-to-normal vision and did not suffer from any psychiatric or neurological disorder except for HFS. The patient sample (*N* = 22, 13 female, 9 male) consisted of 20 right-handed participants and one left-handed participant (one ambidextrous).

All patients of the Movement Disorders Section of the Department of Neurology with blepharospasm and hemifacial spasm were invited to participate in the study provided they met the inclusion criteria. This can most likely be described as convenience sampling.

Both groups were on average approximately 65 years old, with around 13 years of education. The mean disease duration amounted to around 9 years in both groups (for details see Table 1). The Blepharospasm Disability Index (BSDI) [44] and the Jankovic Rating Scale (JRS) [45] reflect limitations of the daily life and physical limitations due to illness. Participants of the study also completed the Montreal Cognitive Assessment (MoCA) [46] in order to detect mild cognitive impairment and dementia. All patients had MoCA scores greater or equal to 24. The Short Form (36) Health Survey (SF-36) [47] was administered as part of the background neuropsychological assessment as a measure of health status; the Beck Depression Inventory-II [48] served as a measure of depressive symptoms. These clinical parameters and further sociodemographic information (including group comparisons) can be extracted from Table 1. All outpatients, i.e., BSP and HFS patients, were treated with local injections of botulinum toxin to suppress motor symptoms. All patients performed the flanker study on the same day when they received fresh botulinum toxin injections. 

Participants received a compensation of 25 €. The ethics committee of Hannover Medical School approved the study (vote number 6589), and all participants gave informed consent in accordance with the Helsinki Declaration. The same patient sample had undergone a Wisconsin Card Sorting Test study aimed at testing cognitive flexibility in our lab [3].

### 2.2. Materials and Procedure

All participants completed 432 trials of a computerized version of the Eriksen flanker task [36] in four blocks after having performed 12 practice trials. Stimuli were presented against a black background by Presentation® (Neurobehavioral Systems, Albany, CA) on a 24” LCD (Eizo EV2416W, Hakusan, Ishikawa, Japan). Stimuli consisted of three white arrowheads which were vertically stacked over one another; the combined size amounted to a visual angle of 4.5° × 1.2° at 1.2 m viewing distance (the vertical distances between the arrowheads were 0.4°). The upper and the lower arrowhead served as distractors (‘flankers’) that could either point in the same direction as the central arrowhead (‘target’) during the congruent condition or point in the opposite direction, which represented the incongruent condition. Flankers appeared 100 ms prior to the target, which was visible for 250 ms, and remained on the screen until target offset. Subjects were supposed to report the central arrowhead’s orientation irrespective of the flankers’ alignment. Responses were collected by a Cedrus® response pad (RB-830, Cedrus, San Pedro, CA, USA). Since the target’s direction could be either pointing left or right, the participants were instructed to press the very left or very right button with their left or right index finger, respectively.

The subsequent trial started 800 ms after the response was given (RSI). The stimulus material is depicted in Figure 1. Possible trial sequences are listed in Table 2. The behavioral data reported here were collected as part of an event-related potential study (unpublished data). The electroencephalogram was recorded from all subjects during completion of the flanker task. The vertical alignment of the arrowheads served to reduce the occurrence of eye movement during recording. 

### 2.3. Data Analysis

Behavioral data were analyzed using IBM SPSS Statistics 24.0. Inferential statistics were carried out with a predefined significance level of α = 0.05 and reported using partial η² as an estimator for effect sizes of ANOVAs. Reaction times (RT) were calculated for each participant under the following conditions: a correct response was given between 100 and 2000 ms after target onset on a trial, which directly followed a correctly answered trial. This prevents artefactual interactions such as post-error slowing [51]. RTs were calculated as the median response latency for each participant and experimental condition. ERs were calculated as the proportion of error trials in relation to all trials of an experimental condition. Experimental conditions were defined by combinations of the following factors: current congruency (congruent versus incongruent), congruency of the preceding trial (congruent versus incongruent), and response sequence (repetition versus alternation). These within-subjects factors were entered into a repeated-measurement mixed ANOVA with the between-subjects factor group (BSP versus HFS). We used Bonferroni correction for post hoc comparisons. Preceding congruency refers to the flanker–target configuration of the directly preceding trial. The ‘repeated response’ level of the factor response sequence means that participants had to use the same hand and key as they used in the preceding trial to respond to the current trial. The ‘alternated response’ level means that they had to respond using the other.

## 3. Results

### 3.1. Demographic and Neuropsychological Variables

Table 1 displays data on the demographic parameters and neuropsychological characteristics of BSP and HFS patients. There were no statistically significant group differences concerning age, sex, years of education, disease duration, BDI, MoCA, or JRS. BSP patients had higher BSDI scores.

### 3.2. Reaction Time

The mean RT and ER for all the distinguishable trial sequences are shown in Table 3. The ANOVA of RT (see Table 4) revealed significant main effects of current congruency—with much longer RTs for incongruent (*M*_I_ = 518, *SEM*_I_ = 9.37) compared to congruent trials (*M*_C_ = 439, *SEM*_C_ = 8.96, *t*(38) = 17.90, *p* < 0.001)—and of response sequence—with longer RTs for repeated (*M*_R_ = 491, *SEM*_R_ = 10.28) compared to alternated responses (*M*_A_ = 467, *SEM*_A_ = 7.99, *t*(38) = 5.13, *p* < 0.001). 

Furthermore, interactions between current congruency and response sequence, as well as preceding congruency and response sequence were statistically significant.

CSE (the two-way interaction between preceding and current congruency) proved significant, indicating that RTs were faster on congruent trials when the preceding trial was congruent rather than when it was incongruent (*M*_cC_ = 432, *SEM*_cC_ = 8.4, *M*_iC_ = 447, *SEM*_iC_ = 9.9, *t*(38) = 3.35, *p* < 0.001), whereas RT on incongruent trials following incongruent trials were faster than on incongruent trials following congruent ones (*M*_iI_ = 514, *SEM*_iI_ = 10.1, *M*_cI_ = 523, *SEM*_cI_ = 8.9, *t*(38) = 2.85, *p* < 0.001). There was no statistically significant group difference concerning CSE (Figure 2a). 

The three-way interaction between preceding congruency, response sequence, and group was significant due to slower RTs for repeated compared to alternated responses following incongruent trials in the BSP group (*M*_iR_ = 500, *SEM*_iR_ = 16.8, *M*_iA_ = 471, *SEM*_iA_ = 12.7, *t*(17) = 3.64, *p* < 0.001), but not in the HFS group (*M*_iR_ = 478, *SEM*_iR_ = 15.2, *M*_iA_ = 472, *SEM*_iA_ = 11.5, *t*(21) = 0.78, *p* = 0.438) (Figure 2c).

### 3.3. Error Rate

The ANOVA of ER (see Table 4) revealed significant main effects of current congruency—with much higher ER for incongruent (*M*_I_ = 0.078, *SEM*_I_ = 0.010) compared to congruent trials (*M*_C_ = 0.013, *SEM*_C_ = 0.003, *t*(38) = 7.41, *p* < 0.001)—and of response sequence—with higher ER for repeated (*M*_R_ = 0.061, *SEM*_R_ = 0.008) compared to alternated responses (*M*_A_ = 0.030, *SEM*_A_ = 0.005, *t*(38) = 5.66, *p* < 0.001). In addition, the preceding congruency showed a significant main effect on the ER with higher ERs for the preceding congruent trials (*M*_c_ = 0.052, *SEM*_c_ = 0.007) compared to the preceding incongruent trials (*M*_i_ = 0.039, *SEM*_i_ = 0.006, *t*(38) = 3.94, *p* < 0.001). 

There were significant two-way interactions between current congruency and preceding congruency (i.e., the CSE), current congruency and response sequence, and preceding congruency and response sequence. The interaction between preceding congruency and group was significant. Post hoc tests showed significantly higher error rates if the preceding trial was congruent compared to incongruent in BSP patients (*t*(17) = 4.08, *p* < 0.001) but not in HFS patients (*t*(21) = 1.36, *p* < 0.181). The ANOVA did not reveal significant group differences concerning CSE (Figure 2a).

The three-way interactions between current congruency, preceding congruency, and response sequence, and between group, current congruency, and response sequence were significant (Figure 2b).

## 4. Discussion

We examined an important facet of executive function, i.e., response inhibition, as well as stimulus and response sequence effects in patients with BSP by means of an objective behavioral RT task. We expected that BSP patients might perform worse than HFS patients on the flanker task, as many studies showed structural and functional abnormalities in BG and BG-related loops in focal dystonia [14,21,22,27], as well as cognitive and psychiatric disturbances in patients with focal dystonia [2,3,4,5,6,7,8]. BSP patients showed stimulus congruency effects and congruency sequence effects that did not differ from those observed in HFS patients. We found longer RTs when responses were repeated after incongruency on the preceding trial in BSP patients compared to HFS patients, suggesting altered mechanisms for repeating responses following incongruency (i.e., cognitive conflict). Confronted with the presence of a cognitive conflict on the preceding trial, BSP patients seem to have difficulties with repeating the previously executed motor response on the current trial. One way to interpret this finding is to consider it as a corollary of impaired habit formation, which enables fast (automatic) responding that does not require controlled processing. It is as though the enduring reverberation of the presence of cognitive conflict interferes in a stronger manner with habitual mechanisms of response selection in BSP patients than it does in HFS patients. 

Previous flanker studies in dystonia have not been published, rendering our study rather exploratory. One limitation of our study is that we did not correct for multiple statistical comparisons for the ANOVA tests included in Table 4, raising the risk for false positives. Thus, our results should be considered preliminary, pending confirmation by follow-up studies. Other study limitations are that a healthy control group was lacking, that BSP patients had greater disability compared to HFS patients, and also that anxiety was not evaluated. We discuss our findings and their limitations in detail below.

Neither the congruency effect nor the CSE differed significantly between BSP patients and HFS patients, and both effects were clearly present in the present data. Hence, our study did not find evidence for a BSP-related modulation of response inhibition or of the sequential adjustment of the spatial extent of attention (when compared to HFS patients). Open questions remain to be delineated because the lack of group differences with regard to both the congruency effect and the CSE may result from truly absent group differences with regard to these cognitive processes, or alternatively, from the rather small-sized samples in our study that compromise the statistical power for detecting such highly specific impairments in task performance.

Earlier research revealed that the CSE was absent in PD patients [43], whereas an effect of the CSE manipulation was observed here in BSP patients and in HFS patients. The previous findings suggested that patients with PD may be characterized by attentional inflexibility, in the sense that they do not adjust the spatial extension of attention as a consequence of cognitive conflict, respectively. However, it should be kept in mind that the absent CSE in patients with PD was detected in comparison to healthy controls. Here, we did not compare BSP patients with healthy controls but with HFS patients, who show similar motor symptoms. Since both PD and focal dystonia (including BSP) are thought to involve different types of BG pathology, further research is needed to examine the differences and similarities among these BG diseases concerning behavioral adjustments. One possibility is that whereas PD is associated with attentional inflexibility, BSP is associated with the kind of response inflexibility that we described above. However, based on the currently available data, this conclusion should be considered as a working hypothesis, which clearly requires confirmatory studies.

RT and ER repetition costs were observed in both groups for current or preceding incongruency. Repetition costs are usually found when the features of the task change across two successive trials [42]. Current stimulus incongruency seems to contribute to repetition costs. In addition, the relatively long RSI of 800 ms may have contributed to the occurrence of repetition costs [40] because RSI that are longer than 500 ms often raise expectations that a response alternation will be required on the upcoming trial [39]. Nieuwenhuis et al. found comparable response repetition costs for incongruent trials, but they did not report inferential statistics [50]. Their RSI was 1000 ms, and they observed modulations of the CSE in response repetition but not in response alternation trials. Pashler and Baylis observed that repetition benefits were attenuated when they increased the RSI from 100 ms to 1000 ms [41]. 

A possible influence of sociodemographic variables, disease duration, and dementia on group-dependent task performance could be minimized by comparing BSP patients with HFS patients instead of healthy controls. Some authors attributed cognitive impairment coinciding with focal dystonia to motor symptoms or pain [9,10,30,31]. An advantageous factor of our HFS control group lies in the between-group similarities concerning motor symptoms, i.e., both groups suffered from the occurrence of eyelid involuntary movements. However, motor symptoms in BSP patients are more severe and more complex than motor symptoms in HFS patients. BSP motor symptoms are also bilateral, thereby inducing stronger disability in instrumental activities of daily living in this group of patients (as revealed by the group differences in BSDI scores). Both patient groups showed depressive symptoms to some degree (Table 1, [52,53,54,55]). Admittedly, we did not measure other psychiatric comorbidities. For example, anxiety levels are typically higher in BSP patients than in HFS patients. Both patient groups were treated with local injections of botulinum toxin at the time of testing. In our view, the discussed group differences should not discourage the recruitment of HFS patients as control group for BSP patients. HFS patients are still considerably more similar to BSP patients than healthy control participants with regard to a number of potentially relevant dimensions. In addition, we think that the specific alteration observed in our study is unlikely to result from differences in general motor functioning (which would rather predict general performance deficits).

The lack of a healthy control group limits the interpretation of our findings to just relative differences between the two neurological conditions, not an absolute difference. This holds true because we do not have a baseline to these groups indicating the deviation direction from normality in the current study. This is another limitation of our study.

Many more studies that are based on narrowly defined and homogenous study groups, suitable paradigms, and adequate sample sizes are required in the field. In our study, BSP patients were examined as a representative sample, but we should consider differences occurring between subgroups of focal dystonia in future research. Previous studies often included phenomenologically heterogeneous groups of patients [29], including focal, regional, and generalized dystonias. This composition resulted in small subgroups, not allowing any valid conclusions about the cognitive differences between them.

With regard to the potential underlying pathophysiological process, we have to admit that we do not have good answers yet. This does not come as a surprise, given that we report behavioral data only. Albeit this suggestion is quite speculative, processes of short-term plasticity might be involved in the generation of our findings: Stimulus-driven habits are considered to involve cortico-basal ganglia pathways [56]. Graybiel [57] suggested that forming these habits depends heavily on experience-dependent plasticity in the BG. She considered dysregulated habit formation as a core symptom in patients with obsessive-compulsive disorder, which in turn is associated with BG dysfunction [58,59] and dystonia [7,60]. Altered BG plasticity is considered relevant for the etiology of dystonia [61]. For example, carriers of the DYT1 dystonia mutation without dystonic symptoms exhibit impaired sequence learning [62], which can be regarded as a symptom of altered BG plasticity. Structural neuroimaging studies point in the direction of BG involvement in the pathophysiology of BSP. In BSP patients [63], bilateral morphometric increases in caudate volume; decreases in putamen volume have been reported as well. The neostriatum (caudate nucleus and putamen) is associated with gradual learning of habits and skills [64], as well as EF [65,66]. According to Jahanshahi and Rothwell [67], any imbalance in fronto-striato-subthalamic-pallidal-thalamo-cortical networks is associated with alterations in habit formation, response inhibition, and the ability to pursue goal-directed actions. Taken together, these data suggest that alterations of BG plasticity play an important role in the pathogenesis of focal dystonia. Our finding of altered response repetition following cognitive conflict possibly may provide a behavioral correlate of some of the neural changes in BSP patients, but clearly, these considerations remain highly speculative. 

## 5. Conclusions

In conclusion, we provide behavioral evidence for altered sequential adjustments of response processing in patients with BSP. More specifically, following cognitive conflict, BSP patients needed more time to repeat the previously executed response than HFS patients. It appears as though incongruency on the preceding trial interfered with a prerequisite of habit formation, i.e., automatic response repetition. Our control group comprised of HFS patients who suffered from comparable, although apparently less severe motor symptoms and from similar comorbidity as BSP patients. Therefore, HFS controls are better suited than healthy controls (who do not show these illness-associated symptoms) for discerning cognitive sequelae of disease-related neural alterations. Considering our findings as a starting point, a promising issue for investigating altered cognition in focal dystonia, in the case that our findings prove replicable, might be impaired habit formation—or other forms of fast routes to action—occurring in conflict-associated situations such as those that follow stimulus incongruency. We considered altered BG plasticity in BSP patients as a possible contributor to an impairment of habit formation. Future behavioral and imaging studies including patients with PD [43] and BSP may provide opportunities to investigate the distinct and specific mechanisms of BG plasticity as they occur in these two groups of patients with probable BG involvement.

## Figures and Tables

**Figure 1 brainsci-10-00076-f001:**
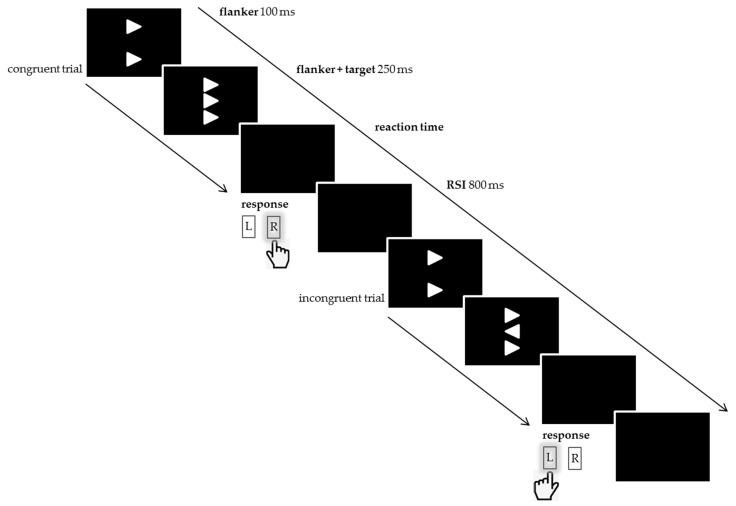
Stimulus materials. This scheme shows two possible consecutive trials, i.e., a congruent stimulus followed by an incongruent one, and the correct responses (R for right and L for left button). Reaction times were individually different. The response stimulus interval (RSI) was 800 ms. Adapted from Seer, C.; Lange, F.; Loens, S.; Wegner, F.; Schrader, C.; Dressler, D.; Dengler, R.; Kopp, B. Dopaminergic modulation of performance monitoring in Parkinson’s disease: An event-related potential study. *Sci. Rep.*
**2017**, *7*, 41222, doi:10.1038/srep41222 [49].

**Figure 2 brainsci-10-00076-f002:**
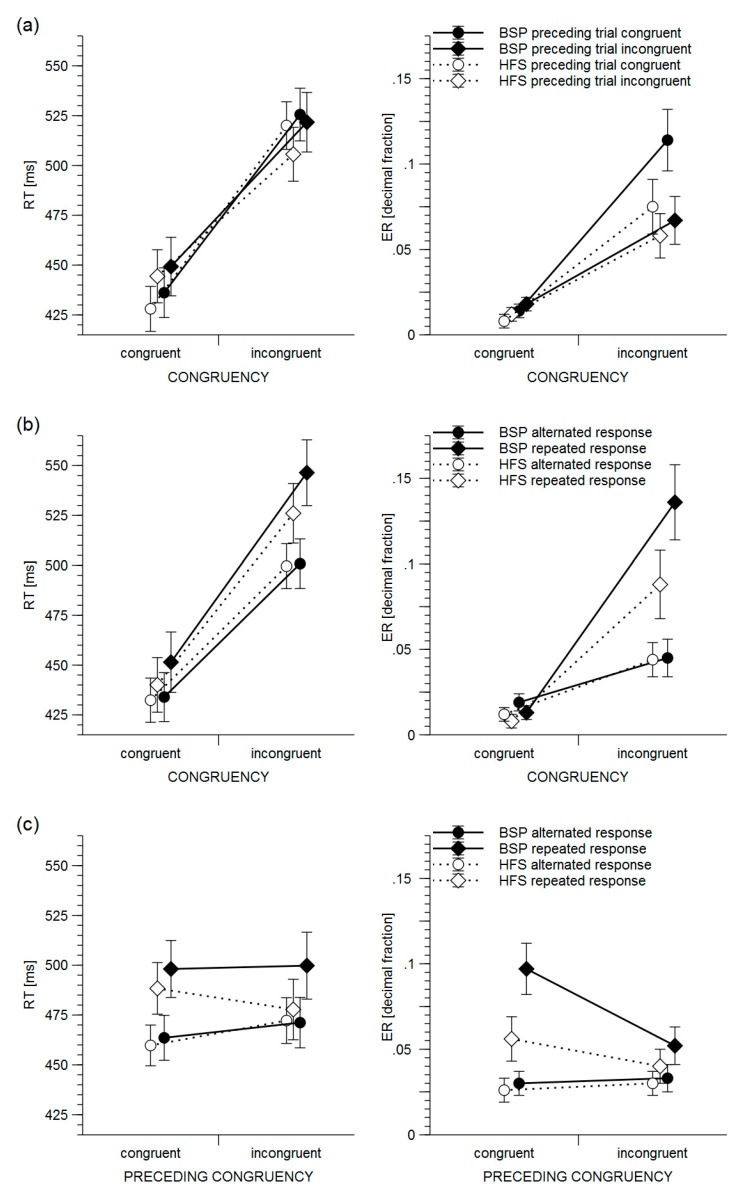
Reaction times (RTs) and error rates (ERs) separately for BSP and HFS patients. Error bars depict standard errors. (**a**) Congruency sequence effect, i.e., the interaction between preceding congruency, current congruency, and group. (**b**) Interaction between current congruency, response sequence, and group. (**c**) Interaction between preceding congruency, response sequence, and group.

**Table 1 brainsci-10-00076-t001:** Sociodemographic data and group comparisons.

	BSP			HFS					
	*N*	*M*	*SD*	*N*	*M*	*SD*	*t*	*df*	*p*
Age (years)	18	65.89	8.55	22	64.17	11.46	0.53	38	0.602
Education (years)	18	12.83	1.86	22	13.23	2.95	−0.49	38	0.626
Disease duration (years)	17	7.88	4.28	17	9.76	6.60	−0.99	32	0.331
MoCA	18	26.94	1.55	22	27.49	1.54	−1.10	38	0.278
	*N*	*M_dn_*	*IQR*	*N*	*M_dn_*	*IQR*	*U*	*Z*	*p*
BSDI	18	1.0	2.00	22	0.1	0.38	79.5	−3.30	< 0.001
Driving	18	3.0	3.00	19	0.0	0.00	44.5	−3.99	< 0.001
Reading	18	1.0	3.00	22	0.0	1.25	144.0	−1.58	0.114
Television	18	1.0	2.25	22	0.0	1.00	120.5	−2.30	0.022
Shopping	18	0.0	2.00	22	0.0	0.00	116.5	−3.03	0.002
Walking about	18	0.5	2.00	22	0.0	0.00	105.5	−3.31	< 0.001
Housework or outside job	18	1.0	2.00	22	0.0	0.00	107.5	−2.97	0.003
JRS	18	3.0	4.25	22	3.0	3.25	169.5	−0.78	0.433
JRS severity	18	2.0	2.25	22	2.0	1.25	159.5	−1.09	0.277
JRS frequency	18	1.0	2.00	22	1.0	2.00	178.0	−0.57	0.567
BDI-II	18	8.5	13.25	22	5.5	9.00	144.5	−1.46	0.144
SF-36	18	70.7	34.53	21	84.1	27.58	132.5	−1.59	0.111
Physical functioning	18	77.5	37.50	21	90.0	25.00	152.5	−1.03	0.301
Physical role function	18	75.0	75.00	21	100.0	62.50	160.5	−0.91	0.365
Bodily pain	18	73.0	51.50	21	100.0	43.00	147.0	−1.24	0.216
General health perception	18	57.0	33.00	21	72.0	26.00	133.0	−1.58	0.113
Vitality	18	52.5	27.50	21	60.0	27.50	134.5	−1.54	0.123
Social role functioning	18	75.0	18.25	21	100.0	6.25	91.0	−2.99	0.003
Emotional role functioning	18	100.0	41.67	21	100.0	0.00	155.0	−1.30	0.194

Welch’s correction was applied for t-tests with unequal variances. Mann–Whitney U-tests were calculated for non-normal distributed variables (IQR: interquartile range). BDI-II: Beck Depression Inventory-II, BSDI: Blepharospasm Disability Index, BSP: blepharospasm, HFS: hemifacial spasm, JRS: Jankovic Rating Scale (measurement for severity and frequency of eyelid involuntary movements), SF-36: Short Form (36) Health Survey, MoCA: Montreal Cognitive Assessment.

**Table 2 brainsci-10-00076-t002:** Possible trial sequences.

Stimulus Array		Trial Type		Repetition (+) vs. Alternation (−)		
Trial *n* − 1	Trial *n*	Trial *n* − 1	Trial *n*	Congruency	Stimulus	Response
► ► ►	► ► ►	c	C	+	+	+
◄ ◄ ◄	► ► ►	c	C	+	−	−
► ► ►	◄ ► ◄	c	I	−	−	+
◄ ◄ ◄	◄ ► ◄	c	I	−	−	−
◄ ► ◄	► ► ►	i	C	−	−	+
► ◄ ►	► ► ►	i	C	−	−	−
◄ ► ◄	◄ ► ◄	i	I	+	+	+
► ◄ ►	◄ ► ◄	i	I	+	−	−

Possible sequences of two consecutive trials characterized by preceding (*n* − 1) and current (*n*) congruency (c/C congruent, i/I incongruent). Note that the same number of mirror-inverted trials—i.e., the target of the current trial points to the left instead of right—is not depicted. Each of the 16 sequences appeared 27 times. Adapted from Nieuwenhuis, S.; Stins, J. F.; Posthuma, D.; Polderman, T. J. C.; Boomsma, D. I.; De Geus, E. J. Accounting for sequential trial effects in the flanker task: Conflict adaptation or associative priming? *Mem. Cogn.*
**2006**, *34*, 1260–1272, doi:10.3758/BF03193270 [50].

**Table 3 brainsci-10-00076-t003:** Mean reaction times (RTs) and error rates (ERs) depending on trial sequence.

Trial *n* − 1	Trial *n*	Response Sequence	RT (*M*(*SD*))		ER (*M*(*SD*))	
			BSP	HFS	BSP	HFS
c	C	+	445(58.3)	435(63.1)	0.010(0.025)	0.005(0.010)
c	C	−	427(46.8)	421(52.0)	0.017(0.031)	0.010(0.019)
c	I	+	551(63.2)	542(67.2)	0.184(0.165)	0.107(0.064)
c	I	−	500(54.3)	498(49.4)	0.044(0.038)	0.042(0.048)
i	C	+	458(75.0)	445(65.4)	0.016(0.031)	0.010(0.013)
i	C	−	440(61.8)	444(56.1)	0.020(0.031)	0.013(0.019)
i	I	+	542(99.4)	511(57.3)	0.088(0.095)	0.069(0.056)
i	I	−	502(56.9)	501(53.7)	0.047(0.061)	0.046(0.049)

Reaction times [ms] and error rates [decimal fraction] depending on preceding (*n* − 1) and current (*n*) congruency (c/C congruent, i/I incongruent) and response sequence (+ repetition, − alternation), separately for groups.

**Table 4 brainsci-10-00076-t004:** Overview of ANOVA results for reaction times (RTs) and error rates (ERs).

Measure	Factor or Interaction	*F*	*p*	η^2^_p_
RT	congruency	320.27	<0.001	0.894
	preceding congruency	0.82	0.371	0.021
	response sequence	26.36	<0.001	0.410
	group	0.23	0.630	0.006
	congruency × preceding congruency	38.51	<0.001	0.503
	congruency × response sequence	18.24	<0.001	0.324
	congruency × group	0.24	0.626	0.006
	preceding congruency × response sequence	16.75	<0.001	0.306
	preceding congruency × group	0.35	0.555	0.009
	response sequence × group	2.33	0.135	0.058
	congruency × preceding congruency × response sequence	2.93	0.095	0.072
	congruency × preceding congruency × group	3.21	0.081	0.078
	congruency × response sequence × group	0.72	0.401	0.019
	preceding congruency × response sequence × group	5.81	0.021	0.133
	congruency × preceding congruency × response sequence × group	0.19	0.663	0.005
ER	congruency	54.97	<0.001	0.591
	preceding congruency	15.54	<0.001	0.290
	response sequence	32.07	<0.001	0.458
	group	1.57	0.218	0.040
	congruency × preceding congruency	23.27	<0.001	0.380
	congruency × response sequence	49.03	<0.001	0.563
	congruency × group	1.12	0.296	0.029
	preceding congruency × response sequence	2.42	<0.001	0.350
	preceding congruency × group	4.46	0.041	0.105
	response sequence × group	4.11	0.050	0.098
	congruency × preceding congruency × response sequence	16.27	<0.001	0.300
	congruency × preceding congruency × group	3.93	0.055	0.094
	congruency × response sequence × group	5.61	0.023	0.129
	preceding congruency × response sequence × group	3.33	0.076	0.081
	congruency × preceding congruency × response sequence × group	2.56	0.118	0.063

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
