# Peer review of "Flanker Task Performance in Isolated Dystonia (Blepharospasm): A Focus on Sequential Effects"

_brainsci, 2020, doi:10.3390/brainsci10020076_

Round 1

Reviewer 1 Report

The authors have adressed all points adequately. I recommend this revised paper for publication. 

Author Response

Thank you for your helpful comments.

Reviewer 2 Report

The authors’ have addressed a number of my concerns, but several responses are unsatisfactory. Some persistent comments/concerns include:

As stated before, the HFS control group is a strength, but the authors’ stating that their design is “optimal” is not accurate and the idea that there is “no better control group” should be tempered. The patient groups have similar motor symptoms, but BSP can be much more complex motorically (blinking rate changes, brief spasms, long spasms, fluttering, apraxia vs. myoclonic twitching of upper and lower facial nerve innervated muscles on one side of the face). BSP is also associated with high rates of anxiety and depression, dystonia spread (possible some spread was missed given only 2 of 19 reported to have spread), and bilateral symptoms that can lead to greater disability.

A healthy control group would provide a baseline to these two neurological conditions as the deviation direction from normality in the current study cannot be not known. That is, the lack of a healthy control group limits interpretation to just relative differences between the groups, not an absolute difference. As such, the authors should either reference prior studies that involved the Flanker task given to healthy controls (e.g. ref 43) and comment on how present results differ from healthy controls or note that this is a limitation in the Discussion.

Stating PD and BSP affect different networks (line 292) and then stating both involve basal ganglia networks (line 300) is inconsistent. Further it is not clear if the authors are trying to suggest that both BSP and HFS have attentional inflexibility, or that neither have attentional flexibility.

The authors need to fix their sentence line 294 as they can only say a difference was not observed between BSP and HFS patients as per above.

Sentence starting line 319 says “both patient groups were treated” at the time of testing and “thus motor symptoms and pain-related processes were alleviated.” This is incompatible with line 128 statement that the testing was done at a time of maximum symptoms. Which was it?

It is confusing to state an “ERP study gave rise to the behavioral data” (line 131) and that the data “were collected as part of an event-related potential study” (Line 159). Is it a separate study or the same study? The authors response comment is that they were “identical” studies, but in the manuscript, it is called a separate study. And again, if it is a separate study, it is more important to note the timing when demographic/baseline data were collected in relation to Flanker testing.

The authors fail to acknowledge that the much greater disability measured in BSP patients is a limitation in the study and could have confounded their results. Rather the authors expound again that HFS is the best available control group to account for motor symptoms. Another limitation that should be noted is that anxiety levels were not measured. BSP patients have known higher background levels of anxiety and this could have impacted the results.

I understand this is an exploratory study and that the Bonferroni correction method is a conservative one. (This method was raised only in response to the authors initially claiming they applied it.) The authors should also realize that there are other approaches to correct for multiple comparisons that are less stringent. As such it is more accurate in line 264 to not say Bonferroni and just say that correction for multiple comparisons was not applied. Also, suggesting that not correcting for multiple comparisons is the standard in the field in the response is a weak argument for not doing so as it has been clearly demonstrated that there is a serious problem with reproducibility in the scientific literature (see Lab Invest 2019:99;1260–1265 for recent discussion).

The authors’ response to their use of plasticity is that they are referencing “short-term behavioral adaptions.” If that is the case, then why volumetric changes from neuroimaging studies (line 276) and “gradual learning of habits and skills” (line 277) included in the Discussion paragraph about plasticity? My prior point about the plasticity story is that there is no discussion of how an underlying pathophysiological process affecting the sensorimotor striatum might impact only a very specific aspect of the Flanker test (RT in responses following incongruent trials). It may be the authors have no good answer and to state this out right is preferred over adding even greater speculation.

Reviewer 3 Report

i am happy with the changes made in the manuscript, following reviewers comments .

Author Response

Thank you for your helpful comments.

This manuscript is a resubmission of an earlier submission. The following is a list of the peer review reports and author responses from that submission.

Round 1

Reviewer 1 Report

Pekrul et al. have written a concise article about reaction time and error rates on a flanker task in patients with blepharospasm and hemifacial spasm. Their arguments are well-supported and the results are certainly scientifically interesting but the relevance of the results for clinical practice and/or patients' daily life remains unclear.

As the authors state in the introduction, the presence and extend of cognitive problems in dystonia patients remains controversial and is probably restricted to very specific cognitive functions. Neuropsychological data are often confounded by motor symptoms in dystonia patients. The choice of control groups (patients with hemifacial spasm) is an elegant solution to this problem.

 My points are as follows:

Title:

Considering that Albanese et al. (2013) provided an update on dystonia classification, which no longer uses the terms primary and secondary dystonia, and the fact that the study concerns patients with blepharospasm, I suggest to replace “primary dystonia” by “blepharospasm” in the tile.

Introduction:

The introduction is well-structured but it lacks more explicit support for the hypothesis why executive dysfunctions are expected in this patients group. Furthermore, the argument would benefit from more elaboration on why these aspects are important for patients’ daily life and/or clinical practice.

Method:

The sample is described in detail but the sampling method is not mentioned (convenience sampling?). Could this have introduced a bias in the sample?

In the group of patients with blepharospasm, there were two patients who also had oromandibular dystonia. Have the authors checked whether the data of these patients were outliers/significantly different from other scores? The additional motor symptoms might influence the task performance.

In table 1, the authors provide detailed information about the sample. Disease duration is only available for 12 out of 18 patients in the blepharospasm group. Why is that?

Also, the BSDI is developed for patients with blepharospasm. Yet, this instrument is also used to assess severity of hemifacial spasm. Is there research available supporting the use of this instrument in other patient groups? Furthermore, the BSDI scores are higher in the group of patients with blepharospasm. Is this due to the fact that the instrument is designed for this specific group and thus might be more sensitive to specific problem? Or is the severity of motor symptoms greater in the sample o patients with blepharospasm compared to hemifacial spasm?  Especially the last point should also be mentioned in the discussion.

Furthermore, data in table 1 are analyzed using t-tests. However, looking at the means and standard deviations and considering the samll group sizes, it is questionable whether the data follow a normal distribution. If not, non-parametric tests should be applied here.  

The inclusion criteria state that patients did not have psychiatric disorders. Yet, in the group of patients with blepharospasm, mean BDI-II scores are rather high with a large standard deviation. It might be informative to provide ranges of scores and/or to check whether there is an outlier on this measure in the group of patients with blepharospasm.  

In order to make an estimation whether or not fatigue might have played a role in the results, it would be informative to include the overall duration of the procedure. If possible, (maybe in supplemental data) the authors can add information about whether reaction time and error rates increase towards the end of the task and if this increase is the same for both patient groups. It might be that patients with blepharospasm experience fatigue faster than patients with hemifacial spasm, due to more severe motor symptoms and/or higher level of depressive symptoms (see above).

Results:

Following my earlier point about the BSDI scores: Have the authors checked whether these scores correlate with the factors used in the ANOVA? And if they do correlate, why are these scores not included as a covariate?

Figure 2 is a great help in understanding the results.  

Discussion

The discussion is clearly structured and provides a good overview over the most important results. In the first five rows of the discussion, references could be added to the arguments.

As mentioned above, I think that the BSDI scores need to be mentioned in the discussion, as they differ between both groups and the influence on the dependent variables is not yet mentioned.

Reviewer 2 Report

The authors present an investigation into response inhibition in 18 BSP patients using a modified Eriksen Flanker test and a group of 22 HFS patients as controls. The authors report no group differences in reaction time or reaction accuracy, and no group differences in congruency effect, congruency sequence effect, and response sequence variables. A difference found was longer reaction times in the responses following an incongruent trial. The authors suggest this finding may represent a specific deficit in BSP possibly related to altered neuroplasticity in the basal ganglia of BSP patients.

A strength of this study is the inclusion of diseased controls. Weaknesses of this study include lack of a healthy control group, lack of control for depressed mood (nearly significantly difference with standard deviation greater than mean in BSP group) or psychiatric differences, and unspecified timing of clinical demographics and botulinum toxin injections with respect to cognitive testing.

While the authors describe the Eriksen flanker task and its measurable responses in detail, there is no description of the neural bases underlying the different measurable responses and no development of a hypothesis of how this test might be expected to be altered in BSP. The comparison with PD further feels contrived and does not appear to support a hypothesis for the present study such as CSE will be reduced in BSP.

The timing of and response from BoNT treatments in both the BSP and HFS groups represents a very important confounder as it could alter the baseline clinical comparisons, including both motor and psychiatric symptoms, and impact the Flanker testing results. Details should be provided about when testing was performed in terms of the injections, and when the data from Table 1 were obtained. It should also be explained why there are so many missing clinical demographic variables. Of note, the significant differences seen across most disability sub-scores raises questions as to whether HFS group is an appropriate motor control group for the BSP patients.

The authors report using Bonferroni correction for multiple comparisons, but it is not evident this was applied looking at the p values provided for the 30 ANOVA tests included in Table 4. Rather it appears the main results are simply using p=0.05 threshold, which would be inadequate for this very large number of comparisons.

Insufficient discussion provided to explain how the single significant result is supported by a basal ganglia plasticity abnormality that does not affect any of the other aspects of response inhibition being tested. The discussion is also limited in the number of prior studies that support the presence of abnormal basal ganglia plasticity in BSP and how plasticity changes (typically described as being increased in isolated dystonia) would manifest in the relatively specific impairment in response inhibition.

If plasticity is proposed as being responsible for the longer reaction times seen following incongruent trials then disease duration may have some impact on findings and need to be adjusted. A limitation in this regard, however, is that for some reason disease duration is missing in 6 BSP patients (and 5 HFS patients).

The discussion also does not include potential impact of brain regions beyond the basal ganglia, such as the anterior cingulate cortex.\

Some minor issues:

Isolated dystonia is preferred term for the type of dystonia being studies (Albanese Mov Disord 2013).

Parkinson disease is not a “frequent comorbidity” of isolated dystonia. These are distinct conditions and dystonia in PD would be considered combined dystonia.

The timing of this study in relation to the event-related study that gave rise to the behavioral data of Table 1 needs to be explained.

Details of the EEG recording occurring during completion of the Flanker task should be provided. Did all subjects undergo EEG measurement?

There is unnecessary duplication between the results (3.2 and 3.3) and Table 4.

Table 3 should include standard deviations rather than standard errors of the mean.

Figure 2 uses previously unused abbreviation for error rates.